# Awareness and Attitudes of Dental Students toward Older Adults in Indonesia

**DOI:** 10.3390/dj10100183

**Published:** 2022-09-29

**Authors:** Anton Rahardjo, Fakhira Hanna Safira Firdaus, Peter Andreas, Yuniardini Septorini Wimardhani, Diah Ayu Maharani

**Affiliations:** 1Department of Preventive and Public Health Dentistry, Faculty of Dentistry, Universitas Indonesia, Jakarta 10430, Indonesia; 2Department of Oral Medicine, Faculty of Dentistry, Universitas Indonesia, Jakarta 10430, Indonesia

**Keywords:** awareness, attitudes, older adults, dental students

## Abstract

In this study, we aimed to analyse the awareness and attitudes of dental students (DS) toward older adults (OAs) in Indonesia. Compromised oral health (OH) among OAs is a significant public health problem that is a global health burden. Furthermore, appropriate education can provide DS with information about strategic and efficient treatments for OAs. An online invitation was provided to every dental student in Indonesia in the third to sixth year of study to answer a web-based questionnaire as a part of a survey on awareness and attitudes (14 questions) towards OAs. From 1288 valid forms submitted, it was evident that most students (87%) expressed a positive attitude toward OAs. Female students showed a more positive attitude compared to males, with an odds ratio of 1.539 (*p* = 0.036), and students who had a connection to an older adult demonstrated a more positive attitude than those who did not (OR = 2.076; *p* < 0.001). The majority of the DS (98%) were conscious of the importance of geriatrics in dentistry, with female students showing more awareness than their male counterparts (OR = 2.553; *p* = 0.033). Positive awareness and attitudes must be accompanied by knowledge so that DS can offer appropriate and effective oral care to geriatric patients.

## 1. Introduction

The OA population is increasing globally; a population census in 2010 showed that Indonesia was among the top five countries in terms of the proportion of OAs, accounting for 13% of the global population [1,2]. Approximately 80% of OAs live in developing countries [2]. Even though the proportion of OAs in Indonesia is lower than in many developed countries, the absolute number is larger [2]. Evidence has shown that the aging of the population contributes to several changes in oral health problems and settings [3,4,5], and that the degree of dependency of the people in this age group can further influence these changes [6,7]. The World Health Organization has issued a “Call for Public Health Action” on the OH of OAs, as it is crucial for healthy aging and has been frequently neglected [8,9]. OH issues among OAs are increasing globally, which requires attention [10].

A previous study described that OAs in Indonesia had poor dental health and had low awareness of OH behaviours [11]. Most of them never visited a dentist, although needs for dental prostheses were apparent. Another study demonstrated that low body mass index was correlated with a poor oral hygiene index and rural residential characteristics among OAs in Indonesia [12]. The need to improve the OH status and OH behaviours of Indonesian elderly people was emphasized. Due to differences in OH problems and settings in this age group, to be able to provide dental treatment, dentists need reliable knowledge, attitudes, and skills [13].

Preparing DS for the changes in the dental treatment needs of OAs is an important part of the curriculum in geriatric dentistry [14]. In Indonesia, teaching related to geriatric dentistry is usually associated with modules in prosthodontics, oral medicine, and preventive dentistry. However, the topics have not been comprehensive and effective. The dental curriculum needs to focus more on education that emphasises the needs of the aging population and on preparing students with the proper cognitive, psychomotor, and affective training in understanding and addressing OA health issues and challenges [14,15]. Some studies have focused on health professionals and their negative attitude toward OAs; others, specific to dental graduates, revealed that they were not ready to provide dental care for OAs [15,16,17,18]. Globally, research has explored DS’ awareness, perceptions, and attitudes toward OAs, to gain information to promote the development of new strategies or models for geriatric dental education for each country [16,19,20,21,22].

Indonesia is the fourth largest country that has a steadily growing number of OA [1]. Despite this, of all the studies that have evaluated the awareness and attitudes of DS toward OAs around the world, Refs. [16,19,20,21,22] none of them have focused specifically on Indonesia. Therefore, this study aimed at assessing the awareness levels and attitudes of Indonesian DS toward OAs.

## 2. Materials and Methods

### 2.1. Participants

Currently, there are 32 dental schools in Indonesia. Two of them are new and, therefore, only had preclinical students. The inclusion criteria of the study respondents were that they all had to be registered dental students in year 3 to 6 from the 30 dental schools in Indonesia. No specific exclusion criteria were set. Estimation of DS who were aware of the importance of geriatrics in dentistry was 80% of the population. The margin of error for the estimate was set at 3%. With a two-sided 95% confidence interval, and an 80% estimated response rate, the required sample size was 1241 DS.

### 2.2. Procedures

The study was approved by the Institutional Research Ethics Committee at the Faculty of Dentistry in Universitas Indonesia (Protocol No. 090700719). Online informed consent was provided by all participants. The survey was sent to all DS at 30 universities in Indonesia, inviting them to participate in this cross-sectional analysis and to voluntarily complete a self-administered questionnaire survey. The questionnaire link was sent to the respective university coordinators, who in turn distributed and monitored the students as they filled the survey forms. A web-based questionnaire was used for its cost-effectiveness, minimal missing values, and higher data completeness rates compared to a paper and pen format.

### 2.3. Measures

A questionnaire was constructed using past research results [20,23] with some changes to suit the study purpose and population. A modified version of the questionnaire was employed based on previous studies to assess attitude [20] and awareness [23]. Nonetheless, knowledge of DS towards OAs was not explored. The actual questionnaire was translated from English to Bahasa Indonesia by an independent bilingual speaker (IBS). The draft was presented to an expert panel formed to assess the validity. The back-translation was conducted by a second IBS and was compared with the actual English version by the same expert member panel to ensure semantic equivalence. Following revisions based on the expert agreement, the face validity of the questionnaire was assessed among five different university students to obtain a clear picture of the clarity and comprehensiveness of the questionnaire. Cronbach’s alpha for the internal consistency was 0.70, which was considered acceptable.

The final questionnaire had a total of 24 questions that were divided into 3 components. The first component collected the participant’s demographic data, including sex (male/female), year of study (year 3–4/year 5–6), stage of study (preclinical/clinical), age (≤20/>21), type of university (public/private), area of university (Java/non-Java), currently living grandparents (yes/no), relationship with grandparents (yes/no/no because my grandmother and grandfather passed away), and connection with any other OAs (yes/no). The second component pertained to the dental student’s awareness of the importance of geriatrics in dentistry using “yes or no” dichotomous categorical responses. The last component collected information on the attitudes of the students toward OAs using 14-item statements with Likert scale responses comprising the following ranking options: 1 denoted “Strongly Disagree”, 2 for “Disagree”, 3 for “Neutral”, 4 for “Agree”, and 5 for “Strongly Agree”. Based on the mean score, the participants were rated to have a negative attitude (score 0–2.99), neutral attitude (score 3.00), or positive attitude (score 3.01–5.00). An analysis was conducted comparing positive attitude with negative and neutral attitude.

### 2.4. Analysis

IBM SPSS Statistics version 23.0 (SPSS Inc., Chicago, IL, USA) was used for data analysis. The data were initially analysed descriptively. Additionally, the chi-square test and logistic regression (LR) were conducted to analyse the comparison and relationship between students’ attitudes toward and awareness of OAs and factors, namely sex, year of study, stage of study, age, type of university, area of the university, currently living grandparents, close relationship with grandparents, and connections with any other OAs. The significance level and confidence interval (CI) were set at 0.05 and 95%, respectively.

## 3. Results

In total, 1288 valid questionnaires were collected, originating from the 30 dental schools in Indonesia that participated in our cross-sectional analysis. Table 1 presents the profiles of the students and the variables associated with awareness and attitudes toward OAs. Most of the students were female (n = 1073, 83.3%); in addition, the majority agreed to the importance of geriatrics in dentistry (n = 1264, 98.1%). Table 2 contains the statements on the attitudes that were asked of the students and the percentage of the responses. Table 3 displays the chi-square analysis between awareness of the DS of OAs and the selected factors. Through the LG, it was shown that sex was a significant factor related to awareness (Table 4). Overall, female DS were more highly aware of the elderly population than male DS with an OR of 2.553 (CI of 1.079–6.044; *p* = 0.033).

Table 5 describes the relationships between attitudes toward OAs and the independent variables. Sex, a close relationship with grandparents, and connection with any other OAs were found to be correlated with the attitudes of the DS. The results of the LG confirmed that sex and the presence of a connection with other OAs were significantly associated with the students’ attitudes (Table 6). The results of the LG confirmed that the female DS had a higher positive attitude toward OA than did male DS (OR = 1.539; 95% CI = 1.029–2.301; *p* = 0.036), and those who had a connection with other OA demonstrated a more positive attitude than those who did not have such a connection (OR = 2.076; 95% CI = 1.495–2.884; *p* < 0.001). Furthermore, no floor or ceiling effects were present, showing that it is unlikely that extreme items are missing in the lower or upper end of the scale, indicating sufficient sensitivity. Nonetheless, test-retest reliability was not conducted in this study.

## 4. Discussion

The strength of this study lies in the fact that it is the first study conducted to assess the awareness and attitudes of Indonesian DS toward OAs with a relatively large sample size. However, this study had some limitations. First, the study tried to include all the DS from all the schools in Indonesia; however, the response rates of the DS who were located outside Java were lower than those of the DS located on Java Island. The low response rate may cause selection bias, affecting the representativeness of the study for overall dental students in Indonesian. A connection with lecturers from schools outside Java to help inform students about this study might have increased the response rate. Second, another limitation of the study may be that the students who participated might be the ones who are interested in gerodontology; therefore, the interpretation of this study’s results should be considered with caution. Although individual variation always occurs and no measurement is perfectly accurate, the sample size of this study was relatively high, which might reduce any potential errors. Third, the cross-sectional design suggests association but cannot support causality. Future longitudinal studies are needed to follow reported awareness and attitudes of dental students toward older adults in Indonesia over time.

Almost all the DS who participated in this study were aware of the importance of geriatrics in dentistry. Sex influenced the awareness of the DS toward OAs, with female students being more aware than male students. This may be due to the greater social sensitivity of females. Groups with more females have exhibited a great sense of equality in conversation that enables the group members to have a healthy discussion among themselves, thereby making the best use of their knowledge and skills [16,24,25]. Moreover, our study revealed that sex was also associated with the attitude of the DS toward OAs. Female DS had a stronger positive attitude than male DS, which aligns with the results of reports from Iran and Brazil [19,20], although they seem to contrast with other studies conducted in the USA, Germany, and India [18,26,27]. Nonetheless, these results must be interpreted with caution due to potential bias. There were an unequal number of males and females. This unbalanced composition reflects the fact that the majority of DS in Indonesia are females [28,29].

The inculcation of optimistic attitudes toward OAs in DS may be vital for establishing a more positive professional behaviour and practice in dentistry [19,26]. It has been shown that personal experience and societal influence determine the positive attitudes toward OAs [27,30]. Furthermore, the development of empathy since childhood and during dental professional education may also influence the results of this study. The intensive clinical exposure of students, with proper guidance, and increased community visits and interaction with people, might bring about the necessary change in the attitudes of the students [27].

In addition, the findings of this study correspond with those of other studies, revealing that the DS who had a connection with other OAs had a more positive attitude toward this age group compared to those who did not [20,31,32] This may be related to increasing familiarity with the aging process and the physical features of old age due to exposure to OAs [20,32]. The behaviour and values of health care professionals can be affected by the conditions of their families or friends, which leads to greater positive attitudes [20,32].

More than 80% of the DS in the study disagreed with the statement that the treatment of older patients with a chronic illness was hopeless. Indeed, research has shown that chronic illness can be controlled through effective behavioural changes, appropriate medical management, and systematic monitoring, and chronic diseases and their consequences can often be prevented or managed effectively [33,34].

Our results also revealed that more than 60% of the DS agreed that they tended to pay higher attention to and sympathize with geriatric patients than those who are younger. This proportion was similar to the one found by a previous study in Iran [20]. In the present study, almost half of the DS responded neutrally to the statement that obtaining a medical history from OA patients is an ordeal. This may be the result of inadequate training and the experience of the students about the delivery of oral health care to OAs [35]. Effective and affordable strategies and programs should be designed to encourage public health care administration to provide better OH and quality of life (QOL) for OAs [8].

Additionally, more than 70% of the students agreed that as people age, they also become less organized and comparatively more confused. They know that aging is associated with physical, cognitive, and functional alterations, especially in patients with dementia, who rarely follow the recommendations of a dentist [35]. The awareness and attitudes of the DS toward the elderly population in Indonesia are positive. However, half of the participants in the current study would still prefer to see younger patients rather than older ones, which is a similar result to a previous study assessing the same issue in dentists [19]. This may be due to the lack of knowledge on how to treat OAs [15,36]. Although, the topic about the theory of the aging process, relation of aging to the condition of the oral cavity, oral problems related to older adults in terms of tooth loss and its treatment, and systemic conditions found in dental patients have been covered throughout the pre-clinical years of the dental curriculum. The assessment of students on these topics were done based on the semester of the pre-clinical years which covered the topics. Studies to clarify the possible factors that influence this attitude in practicing geriatric dentistry are needed.

The finding of the study may be related to the fact that the lack of further information and education may restrict the students from providing strategic and efficient treatments to OAs [37,38]. Thus, the development of a curriculum related to geriatric dentistry to improve positive attitudes and practices shall be provided to DS in Indonesia. A further implication of the result of this survey is that it was conducted with respondents at the undergraduate level. Training in outreach facilities and community settings may improve students’ competencies in managing older patients, including weak and dependent persons [36,38]. Moreover, opportunities to develop postgraduate gerodontology courses are warranted in Indonesia to address the importance of formal training in comprehensive OH care for OAs [39,40]. Continuing education and training opportunities in OH care for OAs that emphasizes interdisciplinary and interprofessional approaches should be increasingly offered. This is indispensable due to the ever-growing number of OAs in Indonesia.

## 5. Conclusions

The awareness and attitudes of Indonesian DS toward OAs have been explored and analysed. The findings of this study confirmed that the DS in Indonesia have a positive awareness and attitude toward OAs. This study showed that sex and the existence of a connection with OAs were significant predictors of increased awareness and positive attitudes among the DS in Indonesia. Female students and students who have a connection with OAs and are more exposed to the process of aging have a more positive attitude toward this population. The positive awareness and attitude found in DS need to be accompanied by knowledge so that DS can offer appropriate OH care to OAs in the future.

## Figures and Tables

**Table 1 dentistry-10-00183-t001:** Profiles of the students and the variables associated with awareness and attitudes toward older adults (n = 1288).

Variables	N (%)
**Demographic information**	
Sex	
Female	1073 (83.3%)
Male	215 (16.7%)
Year of study	
Year 3–4	629 (48.8%)
Year 5–6	659 (51.2%)
Stage of study	
Preclinical	681 (52.9%)
Clinical	607 (47.1%)
Age, year	
≤20	349 (27.1%)
>21	939 (72.9%)
Type of university	
Public University	831 (64.5%)
Private University	457 (35.5%)
Area of university	
Java	648 (50.3%)
Non-Java	640 (49.7%)
**Connection with older adults**	
Have grandparents who are currently alive	
Yes	936 (72.7%)
No	352 (27.3%)
Have a close relationship with grandparents	
Yes	924 (71.7%)
No	97 (7.5%)
No, because my grandmother and grandfather passed away	267 (20.7%)
Have a connection with any other older adults	
Yes	747 (58%)
No	541 (42%)
**Awareness**	
Do you realize the importance of geriatrics in dentistry?	
Yes	1264 (98.1%)
No	24 (1.9%)
**Attitude**	
Positive	1119 (86.9%)
Neutral	62 (4.8%)
Negative	107 (8.3%)

**Table 2 dentistry-10-00183-t002:** The percentage of dental students’ agreement with the statements given on attitudes toward older adults.

Statements	Agreement	Neutral	Disagreement
Most old people are pleasant to be with.	46.0	51.2	2.8
It is society’s responsibility to provide care for its elderly persons.	87.5	10.5	2.0
Old people in general do not contribute much to society.	11.4	35.2	53.4
In general, old people act too slowly for modern society.	24.3	43.4	32.3
It is interesting listening to old people’s accounts of their past experiences.	82.8	14.2	3.0
The government should prioritize allocating money other than to geriatrics.	84.5	13.4	2.1
Medical care for old people uses up too many human and material resources.	15.6	43.2	41.2
Old people do not contribute their fair share towards paying for their health care.	5.1	32.3	62.6
If I have the choice, I would rather see younger patients than elderly ones.	21.8	57.0	21.2
As people grow older, they become less organized and more confused.	74.5	19.3	6.2
Elderly patients tend to be more appreciative of the medical care I provide than younger patients.	52.7	38.8	8.5
Taking a medical history from elderly patients is frequently an ordeal.	14.9	46.0	39.1
I tend to pay more attention and have more sympathy towards my elderly patients than my younger patients.	64.6	29.5	5.9
Treatment of chronically ill old patients is hopeless.	3.0	11.8	85.2

**Table 3 dentistry-10-00183-t003:** Variables related to the dental students’ awareness of older adults.

Variable	Awareness	*p-*Value
**Demographic information**		
Sex		0.027 *
Female	1057 (98.5%)	
Male	207 (96.3%)	
Year of study		0.478
Year 3–4	619 (98.4%)	
Year 5–6	645 (97.9%)	
Stage of study		0.366
Preclinical	671 (98.5%)	
Clinical	593 (97.7%)	
Age, year		0.816
≤20	343 (98.3%)	
>21	921 (98.1%)	
Type of university		0.514
Public University	814 (98%)	
Private University	450 (98.5%)	
Area of university		0.703
Java	635 (98%)	
Non-Java	629 (98.3%)	
**Connection with older adults**		
Have grandparents who are currently alive		0.040 *
Yes	923 (98.6%)	
No	341 (96.9%)	
Have a close relationship with grandparents		0.810
Yes	906 (98.1%)	
No	96 (99%)	
Have a connection with any other older adults		0.652
Yes	732 (98%)	
No	532 (98.3%)	

Chi-square test: * *p* < 0.05.

**Table 4 dentistry-10-00183-t004:** Variables related to the dental students’ awareness of older adults.

Variable	Odds Ratio	95% CI	*p*-Value
Sex			
Female	2.553	1.079–6.044	0.033
Male *			
Constant	25.875		<0.001

Logistic regression: CI = Confidence interval, *** Reference category.

**Table 5 dentistry-10-00183-t005:** Variables related to the dental students’ attitudes toward older adults.

Variable	Positive Attitude n (%)	*p*-Value
**Demographic information**		
Sex		0.047 *
Female	941 (87.7%)	
Male	178 (82.8%)	
Year of study		0.055
Year 3–4	537 (85.4%)	
Year 5–6	582 (88.3%)	
Stage of study		0.154
Preclinical	584 (85.8%)	
Clinical	535 (88.1%)	
Age, year		0.170
≤20	300 (86%)	
>21	819 (87.2%)	
Type of university		0.262
Public University	728 (87.6%)	
Private University	391 (85.6%)	
Area of university		0.087
Java	572 (88.3%)	
Non-Java	547 (85.5%)	
**Connection with older adults**		
Have grandparents who are currently alive		0.992
Yes	813 (86.9%)	
No	306 (86.9%)	
Have a close relationship with grandparents		0.043 *
Yes	815 (88.2%)	
No	77 (79.4%)	
Have a connection with any other older adults		0.001 *
Yes	675 (90.4%)	
No	444 (82.1%)	
**Awareness**		
Do you realize the importance of geriatrics in dentistry?		0.567
Yes	1098 (86.9%)	
No	21 (87.5%)	

Chi-square test: * *p* < 0.05.

**Table 6 dentistry-10-00183-t006:** Variables related to the dental students’ attitudes toward older adults.

Variable	Odds Ratio	95% CI	*p*-Value
Sex			
Female	1.539	1.029–2.301	0.036
Male *			
Have a connection with any other older adults			
Yes	2.076	1.495–2.884	<0.001
No *			
Constant	3.203		<0.001

Logistic regression: CI = Confidence interval, *** Reference category.

## Data Availability

The raw data are available from the authors to any author who wishes to collaborate with us.

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
