# Peer review of "Awareness and Attitudes of Dental Students toward Older Adults in Indonesia"

_dentistry, 2022, doi:10.3390/dj10100183_

Round 1
Reviewer 1 Report
This is an interesting study but the manuscript needs revision before it is appropriate for publication.
The title is a bit confusing to me because it is unclear whether the evaluation is about "knowledge" whether this study is an evaluation of "knowledge and perceptions". I think it is the latter, so I suggest rewording the title
Introduction:
very clear and narration driven to the objective of the study
The below studies will help you to reach the objective clearly.
Shaikh SA, Aldhuwayhi S, Joseph AM, Kumari VV, Khan ARA, Thakare A, Manva MZ, Ustad F, Shaikh S, Mustafa MZ, Mallineni SK. Arabian undergraduates perceptions regarding barriers among the geriatric patients for failing to keep dental appointments: A cross-sectional study. J Popul Ther Clin Pharmacol. 2021 Aug 30;28(1):14-23.
Anehosur GV, Nadiger RK. Evaluation of understanding levels of Indian dental students' knowledge and perceptions regarding older adults. Gerodontology. 2012 Jun;29(2):e1215-21.
Methods” authors need to follow the sequence (1) Participants, (2) Measures, (3) Procedures, and (4) Analysis.
Ref No 18 about dental students and Ref No 21 regarding medical and dental students
Authors should kindly explain
Please describe the questionnaires under the new Measures heading. The authors should describe the origin of each measure, along with relevant information about reliability and validity. This includes: (1) number of items for scales; (2) inter-item reliability (Cronbach’s alpha or Kuder-Richardson) for scale items; (3) response categories; (4) items that were reverse coded; (5) if applicable, a measure of validity, such as face validity or expert agreement (to avoid confusion rewrite the details )
An Analysis section is needed to explicate what sort of tests were used for “comparisons.” Were these chi-square, t-tests, anovas, etc? The response format and coding information is missing from the Measures section so it is unclear to me which of these methods is most appropriate for the present study. Once the authors revise the Methods section (to avoid confusion rewrite the details )
Results:
Explained well
The tables are self-explanatory.
Discussion:
The below studies will help you to establish the discussion
Madunic D, Gavic L, Kovacic I, Vidovic N, Vladislavic J, Tadin A. Dentists' Opinions in Providing Oral Healthcare to Elderly People: A Questionnaire-Based Online Cross-Sectional Survey. Int J Environ Res Public Health. 2021 Mar 22;18(6):3257.
Tahani B, Manesh SS. Knowledge, attitude and practice of dentists toward providing care to the geriatric patients. BMC Geriatr. 2021 Jun 30;21(1):399.
Patil PG, Ueda T, Sakurai K. Influence of early clinical exposure for undergraduate students on self-perception of different aspects of geriatric dental care: Pilot study between two colleges from Japan and India. J Indian Prosthodont Soc. 2016 Jul-Sep;16(3):288-93.
The limitations and strengths of the study should be explained in a detailed way.
Keep recommendations at the end of the discussion.
Revise the conclusion.
Reviewer 2 Report
Dear authors,
Thank you for your effort with the study.
Thank you for the effort with your study.
Since empathy and other skills are developed from childhood, has this been contemplated in any way? Had the students been previously assessed at some point before coming to this academic year? I really think it is difficult to draw conclusions from the study with variables not considered.
Reviewer 3 Report
This study has some potential for geriatric dentistry. Although I have some concerns:
-
What were the inclusion and exclusion criteria?
-
As the questionnaire is self-customized, what was the sensitivity test result for the questionnaire?
-
How was the sample size determined?
-
What is the limitation of this study?
-
For easy understanding, I recommend a concise conclusion.
Round 2
Reviewer 1 Report
All the queries have been addressed
Author Response
Thank you very much for the valuable feedback.
Reviewer 2 Report
Thank you for you changes.
Do you have any way to reduce the risk of bias you assume to exist?
Author Response
Thank you very much for the valuable feedback.
Reviewer comment: Do you have any way to reduce the risk of bias you assume to exist?
Response to Reviewer: The description on how to reduce the risk of bias was addressed in the first paragraph of the discussion section. We wrote:
- A connection with lecturers from schools outside Java, to help inform students about this study, might have increased the response rate.
- Although individual variation always occurs and no measurement is perfectly accurate, the sample size of this study was relatively high which might reduce any potential errors.
- Future longitudinal studies are needed to follow reported awareness and attitudes of dental students toward older adults in Indonesia over time.
Reviewer 3 Report
The author's considered all of my recommendations. I don't have any further comments.
Author Response

(The authors gave the same response as above.)
